# Global Nitrogen and Sulfur Deposition Mapping Using a Measurement-Model Fusion Approach

**Hannah J. Rubin[1], Joshua S. Fu[1,2], Frank Dentener[3], Rui Li[4], Kan Huang[5], Hongbo Fu[5]**

[1]Department of Civil and Environmental Engineering, University of Tennessee, Knoxville, TN, 37996, USA

[2]Computational Earth Science Group, Oak Ridge National Laboratory, Oak Ridge, TN 37831, USA

[3]European Commission, Joint Research Centre, Sipra, Italy

[4]Ministry of Education Key Laboratory for Earth System Modeling, Department of Earth System Science, Tsinghua University, Beijing, 100084, China

[5]Shanghai Key Laboratory of Atmospheric Particle Pollution and Prevention (LAP3), Department of Environmental Science and Engineering, Fudan University, Shanghai, 200433, China

E-mail: jsfu@utk.edu

**Keywords**: Measurement-model fusion, nitrogen deposition, sulphur deposition, HTAP II, ammonia, multiple-model mean

**Abstract**
Global reactive nitrogen (N) deposition has more than tripled since 1860 and is expected to
remain high due to food production and fossil fuel consumption. Global sulfur emissions have
been decreasing worldwide over the last 30 years, but many regions are still experiencing
unhealthily high levels of deposition. We update the 2010 global deposition budget for reactive
nitrogen and sulfur components with new regional wet deposition measurements from Asia,
improving the ensemble results of eleven global chemistry transport models from the second
phase of the United Nation's Task Force on Hemispheric Transport of Air Pollution (HTAP-II).
The observationally adjusted global N deposition budget is 114.5 Tg-N, representing a minor
increase of 1 % from the model-only derived values, and the adjusted global sulfur deposition
budget is 88.9 Tg-S, representing a 6.5% increase from the modelled values, using an
interpolation distance of 2.5 degrees. Regionally, deposition adjustments can be up to ~73% for
nitrogen, and 112% for sulfur. Our study demonstrates that a global measurement-model fusion
approach can improve N and S deposition model estimates at a regional scale, with sufficient
availability of observations, but in large parts of the world, alternative approaches need to be
explored. The analysis presented here represents a step forward toward the World
Meteorological Organization's goal of global fusion products for accurately mapping harmful air
pollution deposition.

**1. Introduction**
Atmospheric nitrogen and sulfur deposition from human activities related to the use of fossils
and land use have significant implications for ecosystem and human health. Elevated levels of
nitrogen and sulfur can lead to eutrophication (Anderson et al., 2008; Heisler et al., 2008),
changes in carbon sequestration (Kicklighter et al., 2019; de Vries et al., 2009; Zhu et al., 2020),
loss of biodiversity (Clark et al., 2013; Dise and Stevens, 2005), and acidification (Bowman et
al., 2008). While sulfur deposition is expected to decrease over the next 80 years (Lamarque et
al., 2013), it will remain a serious hazard in many emerging economies. For instance, sulfur
deposition in East Asia peaked in 2006 (Lu et al., 2010) but is still high enough to be concerning,
especially in natural and semi-natural regions (Doney et al., 2007; Luo et al., 2014).
Oxidized nitrogen ($NO_y$) and reduced nitrogen ($NH_x$), together called reactive nitrogen (Nr), and
oxidized sulfur ($SO_x$) deposition occur as wet and dry processes (Dentener et al., 2006). Wet
deposition is measured at hundreds of locations in Europe, North America, and Asia, but dry
deposition is harder to measure and is often instead derived from ambient concentrations and
modeled deposition velocities (Xu et al., 2015). For example, dry deposition is inferred from
continuous concentration measurements combined with modeled dry deposition velocities at a
few locations in North America (Clean Air Status and Trends Network (CASTNET), 2021) and
Asia (Acid Deposition Monitoring Network in East Asia (EANET), 2021).
The United Nations Economic Commission for Europe's Task Force on Hemispheric Transport
of Air Pollution (HTAP) is an international effort to improve the understanding of air pollution
transport science with emissions models. The second phase of HTAP was launched in 2012. Tan
et al. (2018) used the multi-model mean (MMM) of 11 HTAP II chemistry transport models to
estimate the sulfur and nitrogen deposition budgets for 2010. Significant uncertainty remained
due to a lack of station measurements, especially in East Asia, a large contributor to the overall
budget. Tan et al. (2018) compared Acid Deposition Monitoring Network in East Asia (EANET
(Acid Deposition Monitoring Network in East Asia, 2021)) measurements to the MMM output
but there were very few measurements in East Asia and all were located along the southeastern
coast. In contrast, the highest emissions and modeled deposition were inland and north, making it
challenging to evaluate model performance.
Combining measurements and model estimates in a "measurement-model fusion" (MMF)
approach has the advantage of retaining the broad spatial coverage of models while accurately
matching observations. Generally speaking, MMF takes model estimates of concentrations or
fluxes for a region and modifies them based on in-situ point measurements to force the model
towards the observed values (Labrador et al., 2020). One global MMF approach for wet
deposition combined measurements with HTAP I ensemble model values for 2000-2002 (Vet et
al., 2014) where model estimates filled empty grid cells lacking a 3-year observed mean.
Another MMF approach in North America (Atmospheric Deposition Analysis Generated from
optimal Interpolation from Observations, "ADAGIO") used observed concentrations to adjust
predicted concentrations from the Global Environmental Multiscale-Modelling Air Quality and
Chemistry (GEM-MACH) model (Schwede et al., 2019). Recent work in the US (Schwede and
Lear, 2014; Zhang et al., 2019) incorporates Community Multiscale Air Quality (CMAQ) model
output and precipitation data generated by the Parameter-elevation Regressions on Independent
Slopes Model (PRISM, https://prism.oregonstate.edu/, Accessed: 10/01/22), as well as
observations using inverse distance weighting to create total deposition ("TDep",
https://nadp.slh.wisc.edu/committees/tdep/#tdep-maps) maps that are publicly available.
More details of the MMF approach are described in Fu et al. ( 2022) as they lay out a roadmap
for future work, following the World Meteorological Organization's Global Atmosphere Watch
Program (WMO GAW) and the intended role of the MMF Global Total Atmospheric Deposition
(MMF-GTAD) project. This study updates Tan et al.'s ( 2018) global S and N deposition
budgets using a variation of the TDep methodology (Schwede and Lear, 2014) to merge $NH_x$,
$NO_y$, and $SO_x$ modelled gridded deposition fluxes results with deposition fluxes derived from
observations of $NO_3^-$, $NH_4^+$, and $SO_4^{2-}$ in precipitation and precipitation amounts The main
purpose of our study is to demonstrate the viability of a straightforward but globally applicable
MMF approach, while remaining consistent with previous work that provided datasets for impact
assessments for various communities. This approach is an important intermediate step towards
the WMO's goal of reliable deposition products to aid decision-making. We update the 2010
deposition budgets using MMF to combine the broad spatial coverage of a model with accurate
in-situ measurements.
**2. Data Availability**
**Table 1:** Sources of deposition observations.

| Name | Source | Number of Observation Sites | Region | Value |
|---|---|---|---|---|
| NTN, AIRMoN | NADP | 247 | USA | wet deposition |
| CASTNET | NADP | 84 | USA | dry deposition |
| CAPMoN | NAtChem | 27 | Canada | wet and dry deposition |
| EMEP | EMEP | 86 | Europe | wet deposition |
| China Scientific Study | Li et al. 2019 | 407 | China | wet deposition |

| EANET | EANET | 47 | | East Asia | wet and dry deposition |
| IDAF | INDAAF | 1 | | Niger | wet deposition |


All data are from 2010, reported monthly with sources summarized in Table 1. Wet deposition
measurements ($NO_3^-$, $NH_4^+$, and $SO_4^2$) from the US's National Trends Network (NTN) and
Atmospheric Integrated Research Monitoring Network (AIRMoN) are available through the
National Atmospheric Deposition Program (NADP (National Atmospheric Deposition Program,
2021), http://nadp.slh.wisc.edu/NTN/). Measurements were filtered for completeness and quality,
following Schwede and Lear ( 2014). Sites without a full year of measurements or with quality
tags indicating collection issues were not included, resulting in 247 observations in the US. Dry
deposition generated values are available from the Clean Air Status and Trends Network
(CASTNET, 2021) at 84 locations. CASTNET uses an inferential method to calculate dry
deposition fluxes as a product of surface concentration and modeled dry deposition velocity.
Nitrogen and sulfur wet deposition measurements and dry deposition estimates throughout
Canada are recorded by the Canadian Air and Precipitation Monitoring Network (CAPMoN
(2021) and are available through the National Atmospheric Chemistry (NAtChem) database
(https://donnees.ec.gc.ca/data/air/monitor/). Dry deposition estimates from CAPMoN are
calculated by multiplying atmospheric concentration and deposition velocity. There were 27 sites
with a full year of quality checked data for 2010.
The European Monitoring and Evaluation Programme (EMEP (European Monitoring and
Evaluation Prgrarmme (EMEP), 2021; Tørseth et al., 2012), http://ebas-data.nilu.no/) provides
records of precipitation chemistry ($NO_3^-$, $NH_4^+$, and $SO_4^{2-}$) and precipitation depths for Europe.
There were 86 sites with a full year of quality checked data in 2010.
In China, a multi-year nationwide field study, including some of these NNDMN data, was
compiled by Li et al. ( 2019). Daily $NO_3^-$, $NH_4^+$, and $SO_4^{2-}$ site measurements (in mg/L) were
averaged for 2010 for each of the 407 site locations with complete records by multiplying the
concentration by the precipitation recorded at that same site (in mm) and then aggregating to
produce annual precipitation-weighted deposition (Sirois, 1990). For a wider Asian region,
EANET (Asia Center for Air Pollution Research, 2021, https://www.eanet.asia/) wet and dry
deposition and precipitation data are available at 47 sites.
The International Global Atmospheric Chemistry (IGAC) Deposition of Biogeochemically
Important Trace Species (DEBITS) Africa (IDAF) program (Adon et al., 2010; Galy-Lacaux et
al., 2014) has $NH_4^+$ and $NO_3^-$ precipitation concentrations on the International Network to Study
Deposition and Atmospheric Chemistry in Africa (INDAAF (INDAAF – International Network
to study Deposition and Atmospheric chemistry in AFrica, 2021)) website (https://indaaf.obs-
mip.fr/) for one site in Niger. All measurements were converted to mg-N (or S) /m$^2$/yr.

## 3. Measurement Model Fusion Procedure

Global yearly wet and dry $NO_3^-$, $NH_4^+$, and $SO_4^{2-}$ deposition observations (for wet deposition) or
estimates derived from near-surface concentrations and modelled deposition velocities for dry
deposition) were combined with the respective HTAP II model average grid cell estimates, using
model output interpolated to common 1 degree x 1 degree (1$^o$ x 1$^o$) grid cells (Figure 1). For
example, wet $NO_3^-$ deposition observations are combined with the wet $NO_3^-$ modeled deposition
in the nearest HTAP II MMM grid cell to the observation, where observations exist. Dry
deposition values ($NO_3^-$, $NH_4^+$, and $SO_4^{2-}$) from CASTNET and an inverse-distance weighted 1$^o$
x 1$^o$ gridded dataset was created based on the distance from each observation to the center of the
nearest HTAP II model grid cell. Inverse-distance weighting (IDW) was selected as the most
straight forward to implement method to introduce MMF on a global scale while remaining
consistent with previous work (Schwede and Lear, 2014).
The weighting function was calculated as

$$\left(1 - \frac{distance}{max\ distance}\right)^2 \tag{1}$$

following Schwede and Lear's (Schwede and Lear, 2014) approach for the TDep product, where
"distance" is the distance between the site location and the center of the HTAP II model grid cell
nearest to that sampling site location, within a maximum distance of 2.5$^o$ (approximately 280 km
at middle latitudes). The choice of the maximum distance is a crucial parameter for the inverse
distance weighting method in MMF. Prior analysis (e.g. Tan et al. 2018b) has shown that
gaseous and particulate sulfur and nitrogen emissions can travel several hundreds of kilometers,
before being deposited, although there is likely to be a large variation of transport distances due
to regional differences in chemistry, meteorological conditions, transport patterns and removal
processes. These processes interact with spatially heterogeneous emissions. Since there will not
be a single distance that captures the heterogeneity of all processes at play, we present here a
base case using a 2.5° interpolation distance, and two sensitivity cases reducing the distance to 1°
and increasing it to 5°, respectively. The 5° distance can be seen as an upper limit for the distance
where deposition observations can constrain deposition.  The output values of the weighting
function at each observation location are then multiplied by the observed deposition. For the
center of every HTAP II model grid cell near that site, the modeled deposition is multiplied by 1
minus the value of the weighting function. Consequently, if there are no observations near the
model grid cells, the cell value remains the same. The two grid values ([weighting function times
observed deposition] and [1-weighting function times modeled deposition]) are added together to
give the value of the MMF estimate. This has the effect of modifying the HTAP II grid values
only in locations where there are observations within the maximum interpolation distance.
The MMF gridded surfaces were then summed by species along with the remaining unchanged
HTAP II gridded surfaces that lacked in-situ measurements to create total N and S deposition
gridded surfaces (e.g., the MMF wet and dry $SO_4^-$ gridded surfaces were added to the HTAP II
wet and dry $SO_2$ gridded surfaces to get total S deposition). The MMF wet deposition surfaces
include measurements from Europe, Asia, and North America, and the dry deposition MMF
surfaces include estimates from the USA and Asia (see section 2)

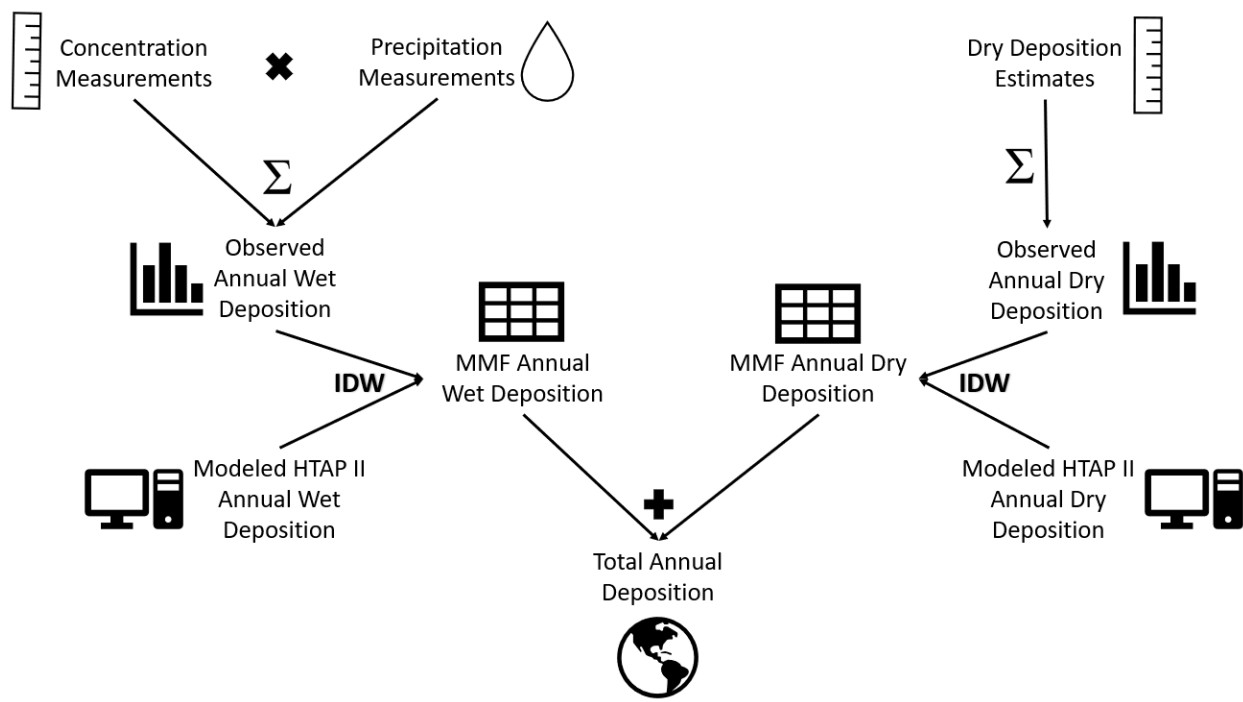


**Figure 1. A flowchart describes the MMF methodology implemented in this paper.**

**4. Results**
The total global $NH_x$ deposition in 2010 increased from 54.0 Tg-N (from HTAP II models) to
54.9 Tg-N (Table 2). Combined with a $NO_y$ deposition of 59.6 Tg-N (from a modeled HTAP II
59.3 Tg-N), the total global deposition is adjusted to 114.5Tg-N (from 113 Tg-N), an increase by
1 %. While the IDW tends to decrease the depositions over the continents, an increase is
calculated over coastal regions and open oceans using the 2.5x2.5 maximum distance. Total S
deposition is adjusted to 88.91 Tg-S (Table 2), an increase by 6.5 % from the HTAP II model
prediction of 83.5 Tg-S (Figure 2B). Regional changes greater than or equal to 10% are bolded
and italicized.

**Table 2: 2010 adjusted global wet and dry deposition in Tg N or Tg S**, MMM indicates Tan et al.'s 2018 multi-
model mean and MMF is this measurement-model fusion work with a 2.5º interpolation distance. The 1º and 5º
interpolation distance results are shown in Tables S1 and S2. Coastal means deposition on sea within 1 degree of the
coastline. RBU is an abbreviation for Russia, Belarus, and Ukraine. Open ocean does not include near-land "coastal"
waters. The regions can be seen in the world map in Figure S1. Regional changes greater than or equal to 10% are
bolded and italicized.

| Region | Non-Coastal | | Coastal | | Non-Coastal | | Coastal | | Non-Coastal | | Coastal | |
|---|---|---|---|---|---|---|---|---|---|---|---|---|
| | MMM | MMF | MMM | MMF | MMM | MMF | MMM | MMF | MMM | MMF | MMM | MMF |
| | **Total NH$_x$** | | | | **Total NO$_y$** | | | | **Total SO$_x$** | | | |
| North America | 3.40 | 3.66 | 0.40 | ***0.31*** | 4.40 | 4.50 | 0.80 | ***0.94*** | 4.70 | **5.67** | 1.30 | ***1.69*** |
| Europe | 2.50 | 2.68 | 0.80 | ***1.14*** | 2.60 | 2.42 | 1.20 | ***1.75*** | 2.70 | 2.50 | 1.50 | ***3.18*** |
| South Asia | 8.60 | 8.60 | 1.00 | 1.00 | 3.60 | 3.60 | 0.70 | 0.70 | 3.70 | 3.70 | 1.00 | 1.00 |
| East Asia | 6.70 | 6.49 | 1.00 | 1.04 | 8.30 | 6.90 | 2.20 | ***2.45*** | 11.20 | 11.89 | 2.90 | ***4.10*** |
| Southeast Asia | 3.20 | ***2.22*** | 1.60 | ***2.12*** | 1.90 | ***1.60*** | 1.40 | 1.44 | 2.40 | ***0.81*** | 2.80 | ***0.56*** |
| Australia | 0.40 | 0.40 | 0.40 | 0.40 | 0.60 | 0.60 | 0.40 | 0.40 | 1.00 | 1.00 | 1.50 | 1.50 |
| North Africa | 0.70 | 0.70 | 0.20 | 0.20 | 1.40 | 1.40 | 0.40 | 0.40 | 1.00 | 1.00 | 0.50 | 0.50 |
| Sub-Saharan Africa | 3.40 | 3.40 | 0.40 | 0.40 | 4.70 | 4.70 | 0.60 | 0.60 | 2.70 | 2.70 | 0.70 | 0.70 |
| Middle East | 0.50 | ***0.38*** | 0.10 | 0.10 | 1.40 | 1.31 | 0.30 | 0.30 | 1.70 | ***3.18*** | 0.60 | 0.60 |
| Central America | 1.40 | 1.40 | 0.60 | 0.60 | 1.20 | 1.20 | 0.80 | 0.80 | 1.40 | 1.40 | 1.40 | 1.40 |
| South America | 3.80 | 3.80 | 0.30 | 0.30 | 3.40 | 3.40 | 0.30 | 0.30 | 2.40 | 2.40 | 0.60 | 0.60 |
| RBU | 1.80 | ***1.18*** | 0.30 | ***0.08*** | 2.40 | ***1.36*** | 0.50 | 0.47 | 3.60 | **5.10** | 0.90 | ***1.17*** |
| Central Asia | 0.50 | ***0.32*** | 0.00 | 0.00 | 0.60 | 0.55 | 0.00 | 0.00 | 1.20 | ***1.88*** | 0.10 | 0.10 |
| Antarctica | 0.10 | 0.10 | 0.00 | 0.00 | 0.10 | 0.10 | 0.00 | 0.00 | 1.40 | 1.40 | 0.00 | 0.00 |
| Continental | 37.00 | 35.33 | 7.10 | 7.69 | 36.70 | 33.64 | 9.70 | 10.55 | 41.00 | 44.63 | 15.60 | ***17.10*** |
| Open Oceans | 9.90 | 11.86 | | | 12.90 | 15.43 | | | 26.90 | 27.18 | | |
| Global | 46.90 | 47.19 | 7.10 | 7.69 | 49.60 | 49.07 | 9.70 | 10.55 | 67.90 | 71.81 | 15.60 | ***17.10*** |


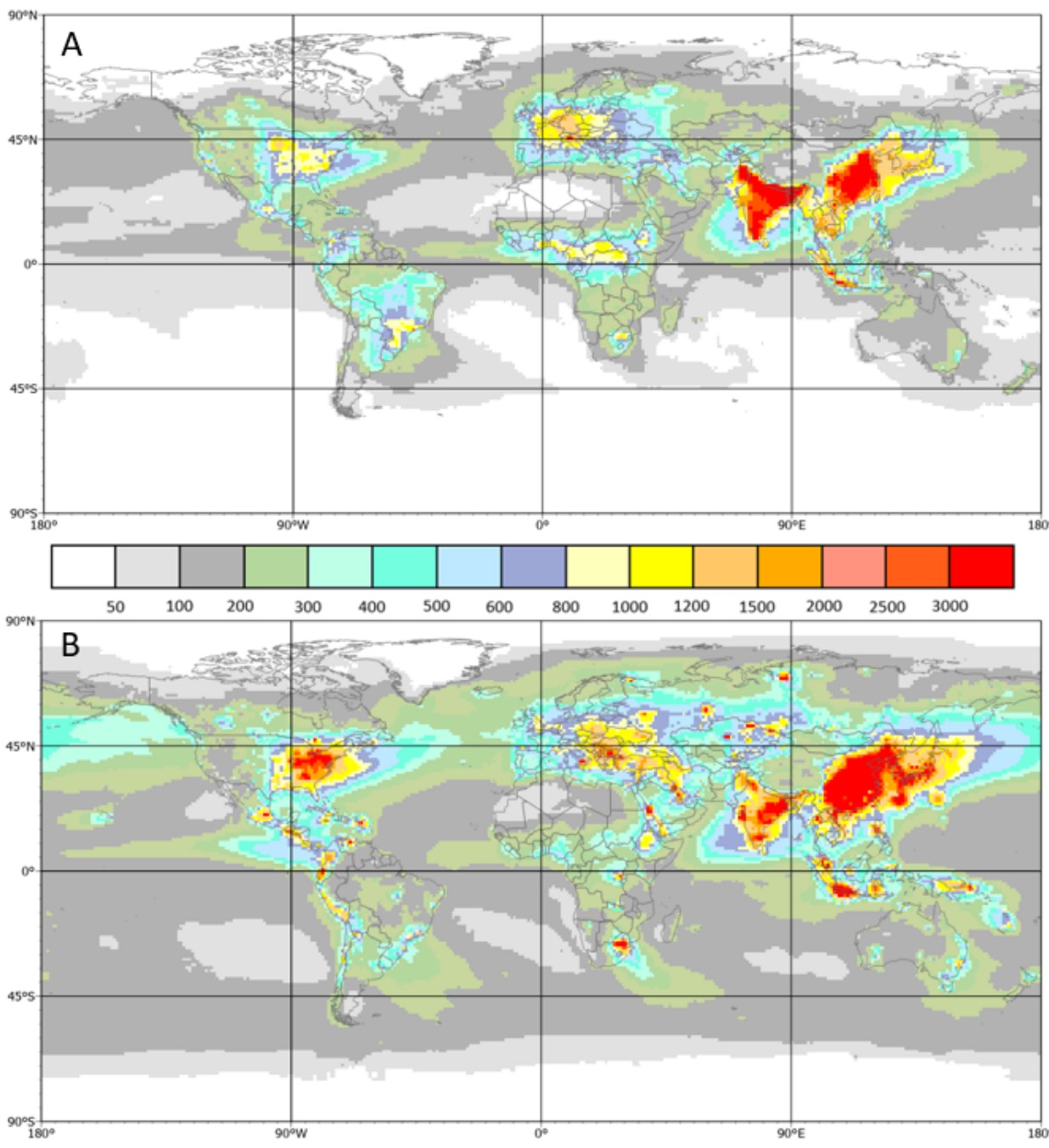

**Figure 2: Total N and S deposition in 2010 using the MMF approach. A)** Total annual N deposition (mg N/m$^2$), the sum of wet and dry NO$_3^-$ and NH$_4^+$ after applying the MMF approach, as well as HTAP II gridded surfaces of dry deposition of NH$_3$, HNO$_3$, and NO$_2$ with no MMF adjustment due to the lack of measurements. **B)** Total S deposition (mg S /m$^2$), the sum of wet and dry MMF SO$_4^{2-}$ and wet and dry HTAP II SO$_2$.

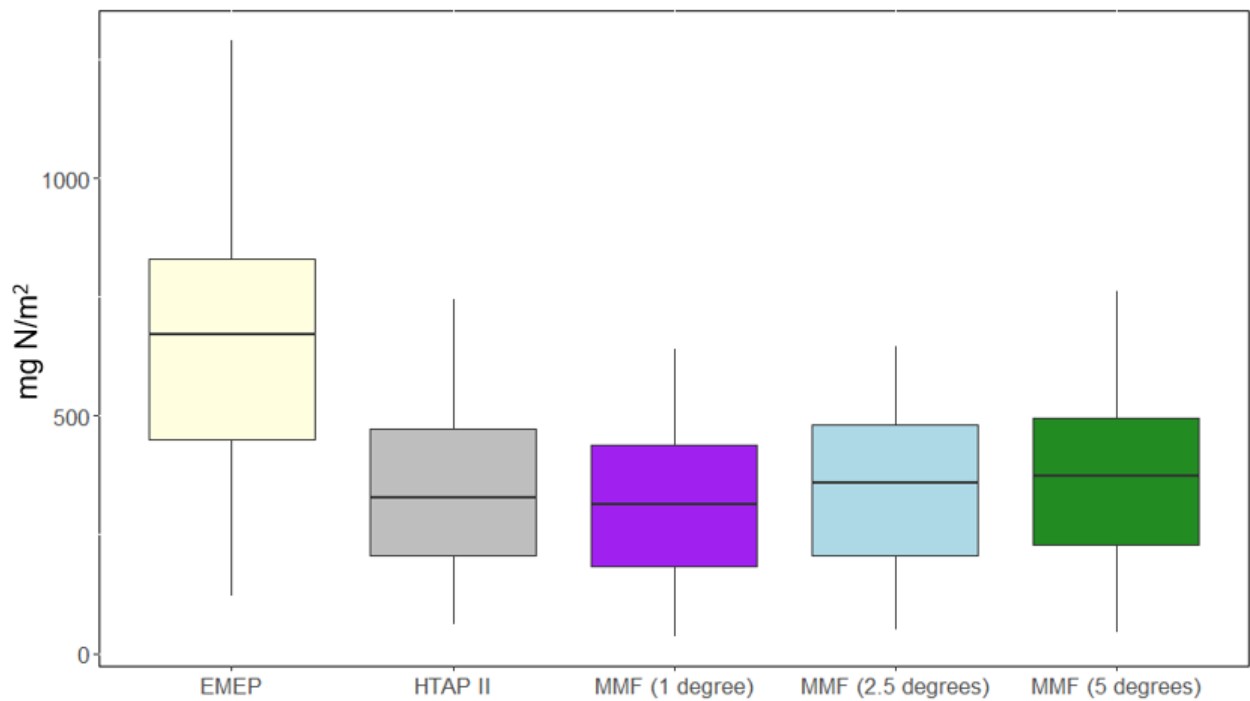

197

**Figure 3: A comparison between HTAP II, MMF, and EMEP wet deposition fluxes in Europe results at**

**EMEP observation sites.** A boxplot shows the distribution of EMEP, HTAP II, and MMF modeled wet reactive

nitrogen deposition ($NH_x$ and $NO_y$) results at each EMEP observation location. Three different interpolation

distances are compared using MMF, 1 degree, 2.5 degrees, and 5 degrees.

Tan et al. (2018) report that their MMM underestimates the high observations of total N

deposition at some EMEP stations in Europe. We find that our 2.5º interpolation value for

European wet N deposition (8.0 Tg) is increased by 12.5% relative to the MMM surface (7.1

Tg), although the distance to the observations remains high (Figure 3). Figures 4, S4 and S5

show the difference between HTAP-II MMM and MMF nitrogen and sulfur deposition in North

America, Europe, and Asia in mg/m$^2$ with different interpolation distances. As the interpolation

distance increases, locations with a single measurement that is very different from the model will

influence the surrounding grid cells to be higher than the model. This effect is in particular

pronounced for sulfur deposition in Southeast Asia (Figure 4 B3) where the MMF procedure

increases deposition by up to 250 mg/m$^2$ relative to the MMM values.

212

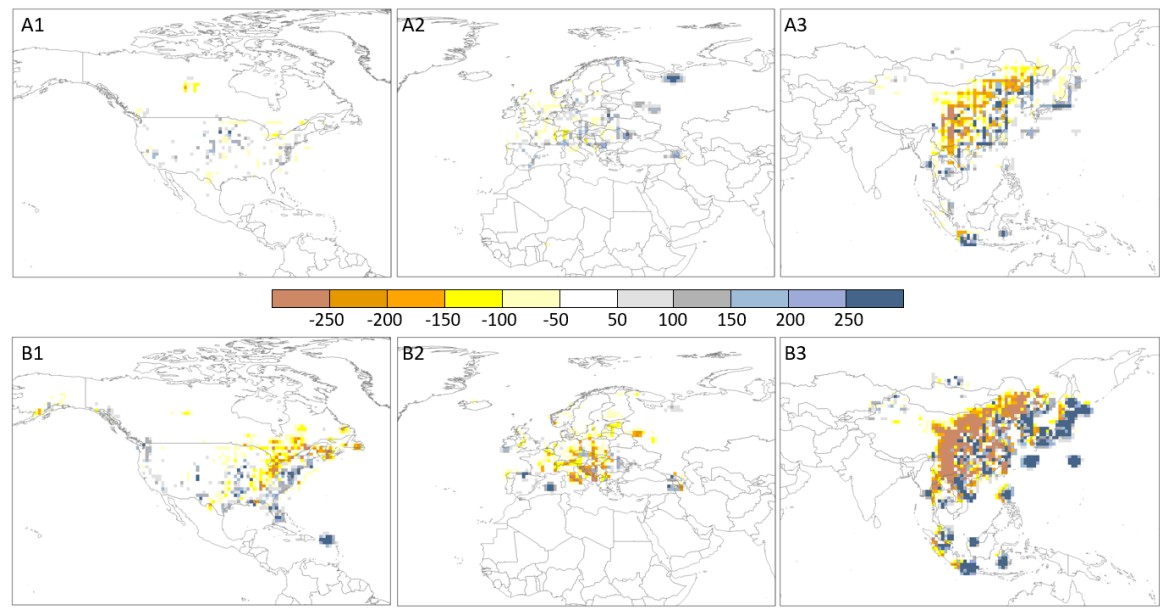

**Figure 4. The difference between MMF and MMM deposition with a 2.5-degree interpolation distance**. **A)** MMF minus MMM reactive nitrogen deposition in North America (**A1**) Europe (**A2**) and East Asia (**A3**) in mg N/m$^2$. **B**) MMF minus MMM sulfate deposition in North America (**B1**) Europe (**B2**) and East Asia (**B3**) in mg S/m$^2$. Results for other interpolation distances are shown in Figures S4 and S5, respectively.

The spatial distribution is slightly different, with more deposition in coastal areas in the MMF estimate (Table 2). Tan et al. ( 2018) report that the HTAP II MMM overestimates $NO_3^-$ wet deposition in North America, but underestimates $NH_4^+$ deposition. We find that the MMF interpolated deposition slightly improves these estimates, although the spatial distribution is very similar with the MMM (Figures 2, 5). The largest change for S deposition (comparing MMM and MMF) is in grid cells classified as ocean because of an increase in East and Southeast Asia deposition which mostly occurs in areas classified as ocean due to the small island size relative to the coarse spatial resolution of the models. We note that, ocean cells were classified as such if they were located further than 1° from the mainland; therefore, any islands smaller than 1° were counted as the ocean.

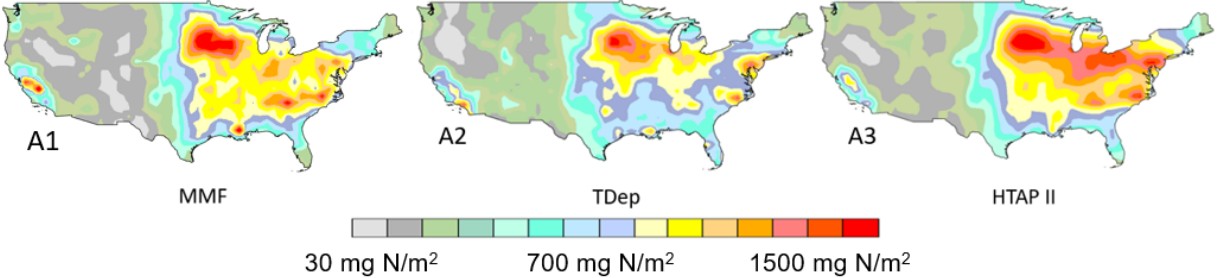

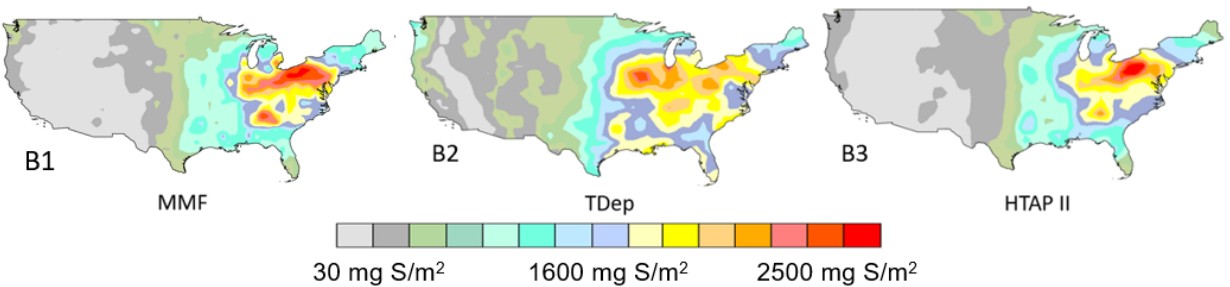

231

**Figure 5: 2010 Total N deposition in the continental USA**. **A**) Total N is modeled with 1) MMF (this work), 2) TDep annual map available from the NADP and 3) Tan et al.'s 2018 MMM. **B**) 2010 $SO_x$ wet deposition in the US as modeled with 1) MMF (this work), 2) TDep annual map available from the NADP, and 3) Tan et al.'s 2018 multi-model mean HTAP II output.

There are spatial differences between an aggregated 1º x 1º version of the original TDep map of nitrogen deposition for the United States as available from the NADP (Figure 5A2), the HTAP II (Figure 5A3) deposition produced by Tan et al. ( 2018) corresponding to the same area, and the deposition map produced in this work (Figure 5A1). A similar pattern is seen in the map of $SO_4^{2-}$ deposition (Figure 5B1; 5B3;5B3). While the TDep maps have been aggregated to the 1x1 degree resolution of the HTAP fields, there is still different regional variation in the deposition patterns in the TDep maps than the HTAP II maps. In particular, TDep is capturing higher west coast values that HTAP II does not while showing lower values in the Midwest/New York/Pennsylvania region.

The $R^2$ value for the linear regression between MMF wet $SO_4^{2-}$ and observed wet $SO_4^{2-}$ in the US is 0.64 (Figure 6). The $R^2$ value for the linear regression between the HTAP II wet $SO_4^{2-}$ and observed $SO_4^{2-}$ is 0.0.60, and 0.89 for the linear regression between the TDep wet $SO_4^{2-}$ and observed $SO_4^{2-}$ (Figure 6). This means that TDep is better reproducing the NADP/NTN

measurements and their spatial differences, whereas the MMF fields remain more similar to the
HTAP II ensemble model output. The higher TDep $R^2$ value likely occurs because of the finer
mesh (12 km) used in the TDep product, the closer proximity to individual stations as compared
to HTAP II used in the MMF approach, and the ability of the regional model to capture
gradients. In principle, emissions should be the same but in global models they are averaged over
larger areas. All three datasets produce similar values to the measured wet $SO_x$ deposition at the
NADP/NTN sites (Figure 6). The $NH_4$ and $NO_3$ wet deposition values are shown in Figures S2
and S3, and have much lower correlations (for all three interpolation distances), with an $R^2$ of 0.1
for $NO_3$ and 0.53 for $NH_4$ at a 2.5° weighted distance

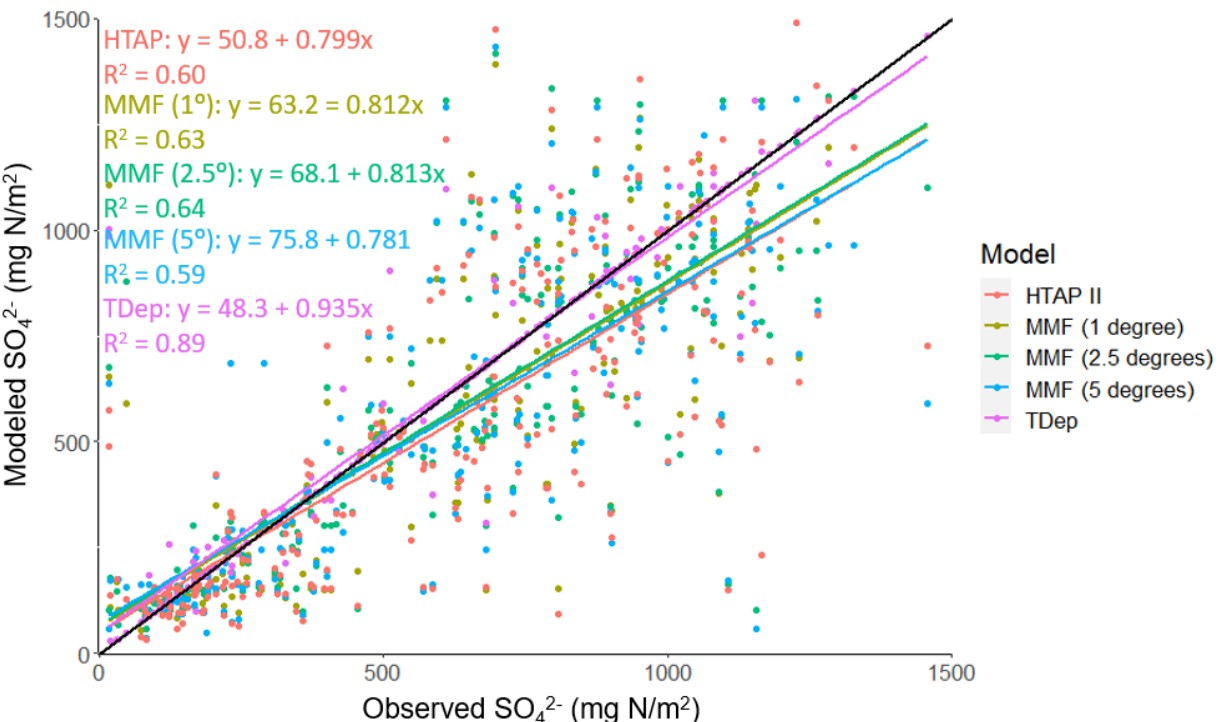


**Figure 6**: **Observed and modeled wet $SO_4^{2-}$ deposition in the US in 2010**. Each NADP/NTN wet deposition
measurement and the associated HTAP II, TDep, or MMF $NH_x$ wet deposition modeled value, with all values
shown together in A. The black line is the 1:1 line. Similar plots are shown in Figures S2 and S3 for wet $NO_3$ and
wet $NH_4$.

**5. Discussion**
*5.1 Consistency of MMF deposition with global emission estimates.*
Geddes et al. ( 2017) used satellite observations to report global $NO_y$ emissions of 57.5 Tg-N/yr
in 2010, similar to the 60.4 Tg-N emissions reported by HTAP II. This matches well with our
total global MMF-derived $NO_y$ deposition (58.1 Tg-N). HTAP II ammonia emissions were 59.3
Tg-N, slightly lower than the MMF $NH_3$ and $NH_4+$ deposition of 62.3 Tg-N. The total MMM
sulfur emissions for 2010 were 90.7 Tg S, very similar to the MMF sulfur deposition of 88.9 Tg-
N.
*5.2 Deposition over China.*
A promising data set of wet deposition measurements ($NO_3^-$, $NH_4^+$, and $SO_4^{2-}$) in China is
available through the National Nitrogen Deposition Monitoring Network (NNDMN (Xu et al.,
2019)). It is comparable to other regional measurements (Wen et al., 2020). However, these data
only exist for a fraction of 2010 (from September onwards) for a few sites; rather than use partial
data to represent an entire year, these sites were not included in our study. Research in China
(Liu et al., 2020) analyzed the spatial pattern of N deposition by combining satellite observations
with NNDMN deposition measurements (Xu et al., 2019); they found a 2012 average of 18.21 kg
N ha$^{-1}$ for China? Additional work combining the GEOS-Chem
(http://acmg.seas.harvard.edu/geos/) model with satellite observations and surface measurements
reports the average annual deposition from 2008-2012 as 16.4 Tg-N with 10.2 Tg-N from $NH_x$
and 6.2 Tg-N from $NO_y$ (Zhao et al., 2017). The averages reported by these studies are consistent
with ours (16.9 kg ha$^{-1}$ yr$^{-1}$) despite the difference in year and spatial resolution. The spatial
pattern of N deposition in 2010 (Figure 2A) also remains similar to that of previous decades (Jia
et al., 2014), with high deposition in eastern China and low deposition over the Tibetan Plateau.
This pattern is confirmed in 2006 and 2013 (Qu et al., 2017).
*5.3 Limitations of interpolation*
As seen in Table 2, the largest difference between MMM and MMF is found in coastal regions
and particularly the open ocean. While MMF does give improved deposition estimates by
incorporating in-situ measurements, it is worth considering the scale of the model. Observations
of deposition are probably not everywhere representative for a 1º or larger resolution and
observations of precipitation may also not be homogenous in all directions at that scale,
especially over  heterogeneous terrain. So, for example, the coarse resolution of the model, even
with added measurements is likely not accurately capturing gradients between coastal and inland
deposition. While higher resolution precipitation values are available in some regions (e.g.,
PRISM in the US), there is still a dearth of both wet and dry deposition measurements. Even on
the North American continental scale, Schwede et al. (2011) showed that partially overlapping
dry deposition estimates from CASTNET (USA) and CAPMoN (Canada)can be very different,
despite using similar methodologies. This adds uncertainty to the dry deposition data (though
there are very few dry deposition estimates included in this study) and emphasizes the
importance of understanding deposition velocity model methodology.
The differences between the TDep, MMM, and MMF gridded deposition (Figure 5) are clearly
visible in the center of the US. While the general patterns of deposition are similar for the three
products, the magnitude of deposition in the aggregated TDep dataset ($1^{\circ}$ x $1^{\circ}$) is higher in the
eastern US and lower in the western US than either of the other two deposition fields. This
difference is likely due to the precipitation dataset used to calculate wet deposition. The MMF
deposition is based on the MMM dataset; therefore, both utilize the same precipitation dataset,
from a combination of 11 global models. However, TDep wet deposition is produced by
multiplying PRISM precipitation data and an interpolated gridded surface dataset of wet $NH_4^+$
concentrations. PRISM is a reanalysis product designed to interpolate precipitation in
particularly complex landscapes using weather radar and rainfall gauge observations, though it is
not identical to observations because it used long-term averages as predictor grids (Zhang et al.,
2018). It captures much more localized variation in precipitation due to geographical variations
which are not captured in the lower resolution global precipitation models used in the HTAP II
MMM (Tan et al., 2018a). To illustrate this, we compare PRISM to the available Community
Atmosphere Model with Chemistry (https://www2.acom.ucar.edu/gcm/cam-chem, "CAM-
Chem"), which was one of the models in the HTAP II ensemble. Subtracting the CAM-Chem
precipitation output over the US from aggregated PRISM precipitation shows that CAM-Chem
greatly underestimates precipitation volume in the US in 2010 (Figure S6). We note, however,
that this comparison does not take differences in precipitation frequency between the model and
observations into account. This matters because if the difference in precipitation volume comes
from a few large magnitude storms, it will not influence the overall wet deposition values much.
This is a good example of the differences that occur when comparing global and regional climate
models and serves to emphasize the importance of resolving spatial and temporal scales. The
total deposition within the US borders is similar for the MMF, HTAP II, and aggregated TDep
gridded surfaces; however, the spatial distribution is different.
MMF and MMM deposition distributions are similar because MMF is based on HTAP II.
Likewise, the MMF results are similar to the TDep values at observation locations because,
despite the difference in precipitation, both utilize the same NADP/NTN measurements to
constrain the models. The key difference between MMF, when compared to MMM, is that
measurement locations are not centered in each $1^o$ x $1^o$ grid cell; therefore, the center of each grid
cell (the value compared to the observation, by interpolation to the station location) will not
exactly equal the measured deposition but will instead be equal to the measurements weighted
proportionally to distance from the centroid. This means that the graphical comparison of Figure
6 is showing the actual measurement locations and 3 different model results with some
meaningful influence from measurements that are nonetheless unique values, except in the very
rare instance that the measurement corresponds exactly to the center of a grid cell. Figure 6
shows a stronger correlation for $SO_4$ than Figures S2 and S3 do for the nitrogen species. This
could be related to the relatively shorter timescales of $NO_y$ and $NH_x$ in the atmosphere. The
relatively coarse resolution of the global models cannot deal with these gradients, so the shorter
timescales are reflected in the observations which are therefore less representative for the larger
grid scales of the models.
TDep maps of North American nitrogen deposition created with Schwede and Lear's
methodology (2014), using IDW, are widely in use and freely available from the NADP. The
sensitivity analysis demonstrates that as the interpolation distance increases, the influence of the
observations on the HTAP II grid increases, smoothing some of the artifacts that can occur using
a small interpolation distance (Figures 6, S2, S3). In this respect it is worth mentioning that the
original TDep dataset for North America used a maximum distance of 30 km plus half the cell
size of PRISM (2.07 km). While it is not entirely clear how this distance was determined,
operational factors such as the station density and the grid size of the regional model are likely
important factors. In contrast, the maximum distances explored in this study are much larger ($1^o$,
$2.5^o$, $5^o$) and are more adapted to the grid size of the current generation of global atmospheric
chemistry transport models, and considerations of transport distances of atmospheric
components. From our analysis there is no obvious better weighting distance that improves the
comparison with observations. An adaptive distance weighting that considers the expected
gradients between the observation point and the remote model grid could be explored as a way
forward.
However, there are strong limitations associated with using IDW (Sahu et al., 2010), and other
interpolation methods such as kriging or geographically weighted regression could provide
smoother surfaces with fewer artifacts. IDW is a fast and flexible interpolation method, but it
does not minimize error and can produce inaccurate results in regions with sparse measurements
and large sub-grid variability. This problem is relevant to much of the world. The lack of
measurement sites globally is a hindrance that can be alleviated by including information
obtained from satellite remote sensing (Walker et al., 2019). Future work should also investigate
methods such as machine learning techniques with spatial information to avoid these limitations.
These results from measurement-model fusion are important because previous methods on a
global scale have relied primarily on models (Vet et al., 2014; Tan et al., 2018a). They compare
their results with measurements, of course, in order to demonstrate the model capabilities but
they do not explicitly incorporate point measurements into the final product. Our results serve to
emphasize that global models are adequately simulating deposition (in terms of total deposition
budgets) but that the regional discrepancies between models and measurements can still be quite
large; and measurement-model fusion helps to ameliorate this without changing the fundamental
model parameters and processes that actually capture the overall deposition reasonably well.
**6. Conclusions**
Sulfur and nitrogen deposition remain a serious concern for human and ecosystem health. We
update the 2010 deposition budgets using measurement-model fusion to combine the broad
spatial coverage of a model with accurate in-situ measurements. The total nitrogen deposition
budget is recalculated to 114.50 Tg-N and the sulfur budget is recalculated to 88.91 Tg-N,
representing about a 1% and 6.5% increase, respectively, from the modelled values. This work
emphasizes the necessity of combining models with observations wherever possible, to better
capture regional patterns and to inform policy and decision-making. Future work to improve
measurement-model fusion should investigate more advanced MMF methods to avoid the
limitations associated with IDW such as surface artifacts and high error in regions with sparse
measurements. It could also incorporate satellite remote sensing derived concentrations to
improve model estimates where in-situ measurements are not available, but a careful error
analysis is needed to avoid spurious results.
**Author Contribution**
HR carried out the methods and analyzed the results. JSF and FD designed the project. HR
prepared the manuscript with contributions from JSF and FD. RL, KH, and HF provided data.
**Competing Interests**
The authors declare no competing interests.
**Code Availability**
Data analysis was done using ArcMap Desktop 10.8.1, ArcGIS Pro, and R (R Core Team, 2022).

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
