# Peer review of "Global Nitrogen and Sulfur Deposition Mapping Using a Measurement-Model Fusion Approach"

_EGUsphere, 2022_

## Author Comment (AC1)

**Response to Reviewers' Comments: Global Nitrogen and Sulfur Deposition Mapping Using a Measurement-Model Fusion Approach**

We thank all the reviewers for their insightful comments. In general, the three reviewers raised a number of methodological issues, all of which are valid in their own right, but also a few that are somewhat beyond the scope of the current manuscript. As now more explicitly mentioned in our introduction, the main purpose of our study is to demonstrate the viability of a straightforward but globally applicable MMF approach while remaining consistent with previous work that provides impact assessments for various communities. As also outlined in our previous paper, the introduction of measurement-model fusion approaches for deposition entails a host of issues and possible approaches, some of which have been demonstrated on regional scales in Europe and North America. However, MMF for deposition has thus far not been applied on global scale, for a variety of reasons, but the most important one is the absence and heterogeneity of available data sources. The World Meteorological Organization (WMO) has recently commenced an activity (MMF-GTAD) that sets the roadmap for improvement of deposition datasets. Our manuscript intends to use a relatively simple method that demonstrates the potential of MMF for Global Nitrogen and Sulfur deposition- a necessary building brick for further progressing the WMO activity. Keeping in mind the overall goal of our paper, we address suggestions individually below.

**Reviewer #1:**

The causes for major revision are: in this work, observed wet deposition is fused with modelled wet deposition. The more common method is to fuse the concentration in precipitation, not wet deposition in itself. Precipitation and wet deposition have a larger variation in space compared to concentration in precipitation. Fusing the latter would allow for a longer length-scale in the fusion. Precipitation should be applied after the fusion of concentration in precipitation.

Response: We thank the reviewer for the comment. While we agree with the reviewer that it in regional studies it is common to fuse the concentration in precipitation, for a consistent approach in our global study this would involve a resource intensive activity of merging precipitation fields and rain concentration fields in global models. Therefore, as a first step, and also to connect to previous global deposition studies, we opted to use a gridded ensemble mean wet deposition dataset created from 11 HTAP-II models. Unfortunately HTAP-II does not provide independent gridded datasets of precipitation and concentration. Within the scope of our current study (see above) we think this is a reasonable method, while the method described by the reviewer will be explored in future work under the WMO MFF-GTAD umbrella.

Here the grid resolution (1 degree) was used as maximum length-scale, which is very short (too short) and will cause for in principle only one or a few grid boxes to be influenced by the observation. There was no scientific explanation to the choice of this length-scale, which can be considered too short. The fusion method explained here, was rather used to estimate the observation error within the gridbox, but is that really how it should be done? Is the observation error dependent on the distance of the observation to the middle of the grid box?

(answer is NO). The grid centre in the model is not the actual centre but an average of the whole grid.

Response: The reviewer raises an important issue. The optimal length scale for inverse distance interpolation of deposition fluxes will depend on multiple factors, including the distribution of the emissions, transformation by atmospheric chemistry, the distribution and intensity of rainfall and the associated removal time scales. The corresponding spatial scales will therefore vary from <100 km to several 100s of kms. We have added a sensitivity analysis where we compare the results of changing the interpolation distance. See Tables S2 and S3 and Figures S2-S4 and 3 (shown below). We fully acknowledge that IDW is not the best measurement-model fusion method, but our aim is to apply globally with all available measurements a method that is already verified and in use officially in the National Atmospheric Deposition Program (NADP) of the US.

[Figure]

**Figure 3: A comparison between HTAP II, MMF, and EMEP results at EMEP observation sites.** A boxplot shows the distribution of EMEP, HTAP II, and MMF modeled wet reactive nitrogen deposition (NH$_x$ and NO$_y$) results at each EMEP observation location. Three different interpolation distances are compared using MMF, 1 degree, 2.5 degrees, and 5 degrees.

The final product shows very little influence from the observations, which is not reasonable to my experience, and a result from the erroneous method. The figures show that the MMM (pure

model) and MMF are very similar also in places with dense observations, and the MMF has a large deviation to observations (see e.g. the bias in deposition in China and US).

The comparison to observations is not clear whether it is by independent or dependent observations, but in whichever case the comparison shows also too weak influence from the observations in the MMF product. It is strange that even in the specific grid box of the observations, the MMF deviate very much to the observed deposition.

For these reasons I recommend that the authors revisit their methods, to improve the MMF results before resubmitting.

Response: While we acknowledge that IDW has some drawbacks, our goal with this paper is to demonstrate that a global measurement-model fusion approach is feasible and can be performed using simple methods. Future work absolutely should investigate other MMF approaches if there is enough or large enough observation data available, but that is beyond the scope of this paper.

We have increased the interpolation distance to 2.5 degrees to address the lack of influence from measurements and added figures to the supplement to demonstrate the impact of either increasing or decreasing this distance. This is now shown in Figure 4, below. And a sensitivity analysis has been added (Figures S4, S5) with a 1 degree and 5 degree interpolation distance. This has changed our results and all totals, tables, and maps have been updated accordingly.

[Figure]

**Figure 4. The difference between MMF and MMM deposition with a 5-degree interpolation distance**. **A)** MMF minus MMM reactive nitrogen deposition in North America (**A1**) Europe (**A2**) and East Asia (**A3**) in mg/m$^2$. **B)** MMF minus MMM sulfate deposition in North America (**B1**) Europe (**B2**) and East Asia (**B3**) in mg/m$^2$.

Row 112-116. I suggest not to include datasets in the manuscript that are only promising. There are likely many promising national datasets in the global arena that potentially could be used, and to include all that are not used will be a paper in itself.

Response: We thank the reviewer for the suggestion.  We understand the concern with a "promising" dataset and it was not included in the study because the data only include a few months of the year 2010, while in following years completely annual coverage is provided. It is now mentioned in the discussion to provide context for the regional dataset we did include and to suggest that future work could incorporate those measurements that are publicly available.

Row 136: title: please state MMF procedure

Response: We thank the review for the comment. We changed the title to "MMF Procedure".

Row 165-166: change to "… include measurements from Asia, Europe and North America, and the dry deposition MMF surface includes measurements from the USA and Asia, …". Explanation: many parts of the world are not covered for wet deposition either. The phrasing was now overenthusiastic about the coverage of wet deposition observations.

Response: We thank the reviewer for the suggestion.  We have changed the phrase as suggested to reflect the lack of worldwide measurements.

Table 1. row open oceans has values in "non-coastal" but not in "coastal". This does not seem correct to me, it should be the other way around. Are the columns mixed up?

Response: The columns are not mixed up; open ocean values are not "coastal" in that they are not near land. A sentence was added to the caption for Table 1 to clarify:

"Open ocean does not include near-land "coastal" waters."

Figures: in general – please label panels a-f etc, it is easier to understand the description if all panels are referred to and labelled.

Response: We have added labels to all graph panels, as suggested.

---

## Author Comment (AC2)

**Reviewer #2**

As a scientific paper instead of data report, however, I have some concerns that need to be further stressed or clarified. Those mainly include the motivation, scientific findings and result evaluation. Details follow.

First of all, I feel the scientific motivation should be better stressed in the Introduction. What is the main purpose of the study? It should be clearly stated. Developing a new method for data fusion, or improving the estimation of global deposition (how to prove it then), or something else?

Response: We thank the reviewer for the comments. We have changed the phrasing at the end of the introduction to explicitly state the goal of the study.

Lines 82-85: "The main purpose of our study is to demonstrate the viability of a straightforward but globally applicable MMF approach while remaining consistent with previous work that provides impact assessments for various communities."

Similarly, could the authors justify their main findings (e.g., the changed estimation of total deposition?) How to demonstrate the numbers were more reasonable compared to existing ones?

Response: We thank the reviewer for the comment. The whole point of MMF is that the numbers will be more reasonable by definition because we are fusing the model estimates toward measured deposition. Tan et al. 2018 describes the lack of agreement between model estimates and measurements in places where there are measurements and we help to fix that by explicitly incorporating measurements to change the model values to more closely match the measured values in places where there are measurements.

Lines 97-100. I can understand that most of dry deposition were obtained based on this method. I am wondering, however, is it possible to collect some dry deposition data from direct observation instead of the inferential method. The latter actually bears some uncertainty from modeled dry deposition velocity.

Response: We thank the reviewer for the comment. Unfortunately, there are only very few direct dry deposition data available worldwide, and the use in a global modelling exercise is very limited. In addition, since our study uses the year 2010 based on the HTAP-II modeling year; we are limited only to data that were collected in 2010 and are publicly available. Of course, in the future, it would be wonderful to have more dry deposition observations.

The procedure part. It is unclear to me whether the authors applied the same IDW method as before, or they made some improvement on the methodology? More importantly, I feel an evaluation on the datasets should be made before conducting the data fusion. For example, how were the observation data compared with simulation applied in this study? Moreover, if

there was big difference between observation and simulation, is it still reasonable to apply the current data fusion method?

Response: We thank the reviewer for the comment. We have not made any improvements to the IDW methodology; but we have changed some aspects of the procedure (we use monthly data rather than weekly). The point of this paper is to demonstrate that existing methods can be applied with existing data on a global scale. There are well-documented downsides to IDW, but we wanted to demonstrate measurement-model fusion with a method that is already routinely in use in the National Atmospheric Deposition Program (NADP) of the US. The observation data were compared with the simulation data and evaluated in previous work (Tan et al., 2018, Li et al., 2019). There is a big difference between observations and the simulated data in some areas; our work seeks to adjust the simulated data to more closely match the observations, and to demonstrate how this influences calculated deposition.

Lines 202-203 (Figure 4). It is quite hard to read "higher observation in Asia are also better reproduced with MMF". Could some quantitative numbers be given?

Response: We have redone our analysis wit three different maximum distances of interpolation and rephrased the sentence to reflect the results.

"Figures 4, S4 and S5 show the difference between HTAP-II MMM and MMF nitrogen and sulfur deposition in North America, Europe, and Asia in mg/m$^2$ with different interpolation distances. As the interpolation distance increases, locations with a single measurement that is very different from the model will influence the surrounding grid cells to be higher than the model. This effect is in particular pronounced for sulfur deposition in Southeast Asia (Figure 4 B3) where the MMF procedure increases deposition by up to 250 mg/m2 relative to the MMM values."

Similarly, lines 226-229. The analysis for the figures are quite descriptive and simple. Can you make more careful comparison and suggest the performance of the three modeling work compared to available measurements?

Response: More details were added to the paragraph describing the figure: "While the TDep maps have been aggregated to the 1x1 degree resolution of the HTAP fields, there is still different regional variation in the deposition patterns in the TDep maps than the HTAP II maps. In particular, TDep is capturing higher west coast values that HTAP II does not while showing lower values in the Midwest/New York/Pennsylvania region."

Figure 6. Why compare wet NH4+ only? It is necessary to provide the comparison for all the species and to make a judgment on data fusion quality.

Response: We thank the reviewer for the comment. The comparison for the other species were added as supplementary Figures 2 and 3.

[Figure]

**Figure S2**. **Observed and modeled wet NH$_4$ deposition in the US in 2010**. Each NADP/NTN wet deposition measurement and the associated HTAP II, TDep, or MMF NH$_4$ wet deposition value. The black line is the 1:1 line.

[Figure]

**Figure S3**. **Observed and modeled wet NO$_3$ deposition in the US in 2010**. Each NADP/NTN wet deposition measurement and the associated HTAP II, TDep, or MMF NO$_3$ wet deposition value. The black line is the 1:1 line.

Line 233-234. Does that mean TDep performed better than this work or the database was more reliable? Then what is the necessity of current work? Should think it over.

Response: Yes, TDep performs better but it is only available for the US.  We are trying to broaden their approach to the rest of the world.

Minor issues:

The title could be quite confusing. "Budget " might not be a proper word as current paper just focused on the deposition.

Response: We thank the reviewer for the comment.  We have changed the title to "Global Nitrogen and Sulfur Deposition Mapping Using a Measurement-Model Fusion Approach" and removed the word "budget."

There is no need to repeat the reference when it is included in a sentence.

Response: We have removed instances where the reference is repeated after being stated in the sentence.

Figure 3: The x-axis and y-axis should be clearly labeled.

Response: We have added letter labels to all graphs.

The language should be improved. Some clauses were not well organized.

Response:  We have tried to improve the language where possible.

---

## Author Comment (AC3)

Reviewer #3:

Independent evaluation: We find many statements of MMM status by HTAP2 within this manuscript; however, the independent evaluation supported by other measurements can increase the persuasion of this MMF result. I agree that MMF will be better in theory but we do not lead the proof that "MMF does give better deposition estimates by incorporating in-situ measurements" (P14, L262-263) without independent validations.

Response: There is no way to do a spatially overlapping comparison, Figures 4, S4, and S5 do clearly demonstrate the impact of MMF on the model estimates. By definition, MMF values will be closer to the measured values because they explicitly incorporate the measured values along with the model estimates.

Not machine learning, but show with test/training site

Response: We thank the reviewer for the comment. As you state, we are not using a machine learning methodology. The common machine learning methods require a large dataset for training and testing. Training and testing sites are not applicable in this case because each measurement is influencing the model to force it towards the measured value. A testing site could be chosen that is influenced by one or several training sites, but taking out the testing site from the full dataset will influence the results. IDW does not build a linear regression or other relationship between inputs and outputs the way machine learning does; it is simply adjusting the values nearby observations.

Already in use in EPA

Response: Yes, exactly, that is why we are applying this method globally. It is already commonly used within the US.

The largest change over the ocean: I can follow that one of the reasons for coarse grid resolution will lead to the largest changes over oceans as listed in Table 1. However, because this was not helped by the observational fact (e.g., ship-borne measurements), how can we interpret this MMF result? Is it possible to only focus on the grid where the observation was available within the 1 by 1 grid in Table 1?

Response: We thank the reviewer for the comment. We have redone the calculations with a 1, degree, 2.5 degree, and 5 degree interpolation distance. As such the results have changed and we expect there to be an increase over the ocean with the larger interpolation distances. This is because some observations are either in grid cells classified as ocean or coastal or are influencing ocean or coastal grid cells.

**Specific comments**

P2, L18-19 (Abstract): Why sulfur trends were not stated? Moreover, according to my major concerns, please rewrite this abstract. It should be clarified the validation of this MMF result.

Response: Sulfur trends are now stated more explicitly in the abstract and the results sections. We cannot have an independent evaluation; this is not a machine learning approach.

P2, L45: It is ambiguous what "it" indicates. Is it ambient concentration or dry deposition?

Response: "It" refers to dry deposition and the sentence has been updated to reflect that.

Line 45: "Dry deposition is inferred from continuous measurements combined with modeled dry deposition velocities…"

P3, L55: No need for the repetition of EANET.

Response: Thanks for the comment. The reference has been changed.

P4, L78-79: I noticed Tan et al.'s paper is updated recently (http://dx.doi.org/10.1016/j.scitotenv.2022.158007). What are the differences between this update and this study?

Response: Tan et al.'s newest paper focuses only on China and only on wet nitrogen deposition.

P4, L86-88: This sentence is the result and is not appropriate to be stated in this introduction section.

Response: The sentence has been removed.

P4, L89: How about preparing table summarization for these available datasets? It will be kind to wide readers.

Response: We have added a new table as Table 1 to summarize the datasets.

**Table 1:** Sources of deposition observations.

| Name | Source | Number of Observations | Region | Value |
|------|--------|------------------------|--------|-------|
| NTN, AIRMoN | NADP | 247 | USA | wet deposition |
| CASTNET | NADP | 84 | USA | dry deposition |
| CAPMoN | NAtChem | 27 | Canada | wet and dry deposition |

| EMEP | EMEP | 86 | Europe | wet deposition |
|------|------|-----|--------|----------------|
| Li et al. Study | Li et al. 2019 | 407 | China | wet deposition |
| EANET | EANET | 47 | East Asia | wet and dry deposition |
| IDAF | INDAAF | 1 | Niger | wet deposition |

P6, L137: It is one of an approach to use wet deposition itself, but their elements (concentration in precipitation and precipitation amount) could be the target of MMF. I can see some relevant discussion in Section 5, but for example, the project of MICS-Asia used the fusion for monthly-accumulated precipitation (https://doi.org/10.5194/acp-21-8709-2021). It will be better for readers why wet deposition is targeted as MMF in this study.

Response: We thank the reviewer for the comment. While we agree with the reviewer that it is common to fuse the concentration in precipitation, we are using an ensemble mean wet deposition grid created from 11 HTAP-II models and therefore we do not have independent precipitation and concentration grids. It is not reasonable to follow the method the reviewer describes given the constraints of our datasets.

P8, L183 (Table 1): It is kind to provide the region map for this analysis as a supplemental figure.

The region map is provided as a supplemental figure (SF1), adapted from Tan et al. 2018.

[Figure]

**Figure S1.** A world map showing the regions used to calculate the totals presented in Table 1. This figure is adapted from Tan et al., 2018, based on their region divisions.

P9, L189 (Figure 2): How about presenting the difference between MMF and MMM to clarify the difference driven by data fusion in this study? This result will clarify the impact of MMF compared to MMM and can help to understand the result listed in Table 1.

Response: We have changed Figure 4 and added figures S4 and S5 to present the different between MMM and MMF to demonstrate the differences driven by the data fusion method.

P10, L202 (Figure 3): But MMF used EMEP dataset itself, so this kind of comparison seems to be meaningless.

Response: Yes, it is true that MMF uses the EMEP dataset; Figure 3 shows to what extent IDW can "correct" the model to match the EMEP results. We cannot have a testing/training split dataset because we are not doing machine learning and there is no model to apply to testing data; therefore, we are limited to looking at the distribution and characteristics of the EMEP and the MMF data points.

P10, L202-203: I do not follow this sentence for East Asia. From this figure, MMF still underestimated the observational values.

Response: Yes, MMF still underestimates the observational values because it is effectively a weighted "average" between observations and model estimates. Therefore, it can correct the model where there are measurements, but the rest of the region's results are based solely on the model. So even "nudging" the model estimates in some grid cells toward the observations is not enough to fully correct the model at those places or over the entire region.

P12, L226-229: Within this context, TDep is regarded as truth?

Response: Yes, that is correct. TDep is widely in use and is created using CMAQ estimates and PRISM precipitation reanalysis and observations. There is a whole team working just on TDep and it has been endorsed by the scientific community.

P12, L230: Why NH4 is only presented? In addition, because MMF uses NADP dataset itself, what is the meaning of this kind of evaluation?

Response: We thank the reviewer for the comment. We now include all species in either the main text or the supplemental figures.

**Technical corrections**

P7, L169: In this Figure 1, "concentration in precipitation" multiplied by "precipitation" should be "wet deposition"? Please confirm this illustration.

Response: Correct. The Figure has been updated to reflect the change of phrase.

[Figure]

**Figure 1**. A flowchart describes the MMF methodology implemented in this paper.

---

## Author Response (AR2)

**Response to Reviewers' Comments: Global Nitrogen and Sulfur Deposition Mapping Using a Measurement-Model Fusion Approach**

**Reviewer #1:**

General comments:

This manuscript is the revised version to address comments raised by three reviewers. The manuscript has been improved; however, I can still find some concerning issues before the possible publication. First of all, I do not raise the wording "machine learning" but used it as I mentioned. I do not understand these responses and intentions. I have again read this manuscript and am still concerned about the following points (most of them will be minor/technical ones but these should be properly done if this manuscript is published by a high-quality journal). I would like to recommend fully checking the manuscript quality before the re-submission.

We thank the reviewer for the comments and we address the suggestions individually below.

Specific comments:

Abstract: I have raised one question about sulfur trends in the abstract and the authors replied "Response: Sulfur trends are now stated more explicitly in the abstract and the results sections. We cannot have an independent evaluation; this is not a machine learning approach.". Without any short introduction to sulfur, this abstract is not well organized. We do not follow why sulfur was analyzed by this method. I would like to request to revise again.

Thank you; this is a good point about the organization of the abstract. We have added an introductory sentence to state the global sulfur trends before we start to describe the methodology.

"Global reactive nitrogen (N) deposition has more than tripled since 1860 and is expected to remain high due to food production and fossil fuel consumption. Global sulfur emissions have been decreasing worldwide over the last 30 years, but many regions are still experiencing unhealthily high levels of deposition. We update the 2010 global deposition budget…"

P2, L45: Is "Nosy" typo?

Yes, it was a typo. It should be $NO_y$.

P4, Table 1: The wording "number of observations" seems to be misread. I guess this should be "number of observation sites".

"Number of observations" has been changed to "number of observation sites."

P8, Table 2: I can find a similar table in the supplemental material (Table S2). This is not mentioned in the main text, and what is the difference between this Table 2 and Table S2? In addition to this point, some supplemental figures/tables were not appropriately referred to in the main text. If the authors prepared the supplemental, it is better to clearly mention it in the

main text. We do not fully capture the supplemental information. Please carefully check the revision process.

Table S1 and Table S2 are now referenced in the text:

"**Table 2: 2010 adjusted global wet and dry deposition in Tg N or Tg S**, MMM indicates Tan et al.'s 2018 multi-model mean and MMF is this measurement-model fusion work with a 2.5° interpolation distance. The 1° and 5° interpolation distance results are shown in Tables S1 and S2."

All other supplemental tables and figures were already referenced in the main text.

P11, Figure 4: Are these figures not using "mg N/m2" and "mg S/m2"? Why the unit is changed?

That is correct – the figures are using "mg N/m2" and "mg S/m2." The description has been updated to reflect the proper units:

"**Figure 4. The difference between MMF and MMM deposition with a 2.5-degree interpolation distance**. **A)** MMF minus MMM reactive nitrogen deposition in North America (**A1**) Europe (**A2**) and East Asia (**A3**) in mg N/m$^2$. **B)** MMF minus MMM sulfate deposition in North America (**B1**) Europe (**B2**) and East Asia (**B3**) in mg S/m$^2$. Results for other interpolation distances are shown in Figures S4 and S5, respectively."

P12, Figure 5: These color scales are not understandable. The maximum value for the upper panel is 1500 whereas for the bottom panel is 2500. However, the medium value is respectively 700 and 600 even though the minimum value is the same. Please carefully confirm this figure.

There was a typo in the color scale. The median value for the bottom panel is 1600, not 600. The units have also been corrected with a superscript.

Technical corrections:

P2, L36-37: Why to start a new line here?

The extra space has been removed.

P6, L129: Only one sentence?

The sentence has been added to the previous paragraph.

P10, Figure 3: Appropriate super-script for the unit.

The super-script has been modified.

P13, Figure 6: Appropriate chemical species name and super-script for the statistical value.

The figure has been updated to have the appropriate species name and super-scripts. The corresponding supplemental figures (Figures S2 and S3) have also been modified.

P14, L284: Maybe no need to use a dot.

The dots have been removed.

"$(16.9 \ \text{kg ha}^{-1} \ \text{yr}^{-1})$"